# Chemical Properties and Microbial Analysis of Waterlogged Archaeological Wood from the Nanhai No. 1 Shipwreck

**Yeqing Han [1,†], Jing Du [2,†], Xinduo Huang [1], Kaixuan Ma [1], Yu Wang [1], Peifeng Guo [1], Naisheng Li [2], Zhiguo Zhang [2] and Jiao Pan [1,\*]**

1   Ministry of Education Key Laboratory of Molecular Microbiology and Technology, Department of Microbiology, College of Life Sciences, Nankai University, Tianjin 300071, China; 2120191000@mail.nankai.edu.cn (Y.H.); 2120191084@mail.nankai.edu.cn (X.H.); makaixuan16@163.com (K.M.); 2120201036@mail.nankai.edu.cn (Y.W.); Gary19981114@163.com (P.G.)

2   National Center of Archaeology, Beijing 100020, China; ldusts@163.com (J.D.); lineas@126.com (N.L.); zzgwys@126.com (Z.Z.)

\*   Correspondence: panjiaonk@nankai.edu.cn

†   These authors contribute equal to this work.

**Abstract:** The Nanhai No. 1 was a wooden merchant ship of the Southern Song Dynasty, which wrecked and sank in the South China Sea, Yangjiang City, Guangdong Province, China. The Nanhai No. 1 shipwreck was salvaged as a whole in 2007 and began to be excavated in 2013. During the archaeology excavation, some of the hull wood fell off the hull. These waterlogged archaeological woods (WAW) were immersed in the buffer containing EDTA-2Na and isothiazolinone K100 for moisture stabilization, preliminary desalination, and microbial inhibition. We evaluated the properties of these WAW through testing the chemical components (including lignin, holocellulose, and ash content) of the wood, and monitoring the iron element content, anion and cation content in the buffer. At the same time, the microbial composition in the desalination buffer was also detected. The results showed that the holocellulose content in these WAW were much lower than in fresh wood. The ash content in these WAW decreased after desalination treatment. The iron element content, anion and cation content in the buffer were high and kept at a certain level after desalination treatment. At the same time, the problem of biodegradation in the buffer should be paid attention to. The comprehensive protection of WAW requires to combine wood properties and microbial problems. This study provides a reference for the protection of WAW from the Nanhai No. 1 shipwreck and other similar historical wood.

**Keywords:** Nanhai No. 1 shipwreck; waterlogged archaeological wood (WAW); wood properties; microbial composition

## 1. Introduction

The Nanhai No. 1 shipwreck was a wooden merchant ship of the Southern Song Dynasty (1127 to 1279 AD), which sank in the South China Sea in Guangdong province, China. After the Nanhai No. 1 shipwreck was salvaged as a whole out of the sea, it was placed in the Maritime Silk Road Museum [1]. Many precious cultural relics had been unearthed from the Nanhai No. 1 shipwreck. Among all the cultural relics from the Nanhai No. 1 shipwreck, the wooden hull cultural relic was the most precious and the most difficult to protect, not only because the Nanhai No. 1 shipwreck is huge in volume, but also because the corrosion degree of the hull was uneven. The excavation of the wooden hull cultural relics ran through the excavation of the Nanhai No. 1 shipwreck since 2013. Maritime wooden cultural relics had been immersed in seawater for a long time, and their preservation environment was closed, at a constant temperature, constant pressure, high salt, and oxygen deficient environment. When they were excavated, the preservation environment changed into an open environment where the temperature,

humidity and air circulation were not easy to control. Moreover, the control of oxidation, corrosion, microorganism biodegradation and other diseases would become extremely difficult. In order to maintain the stability of wooden cultural relics and control or delay the breeding and development of diseases, protection measures such as cleaning, moisturizing, desalination, reinforcement and anticorrosion and so on are usually taken. A series of scientific on-site protection measures laid the foundation for the transition from on-site protection of cultural relics to laboratory protection and restoration. During the excavation of the Nanhai No. 1 shipwreck, a large number of scattered individual pieces of wood were unearthed. After excavation, this hull wood was usually immersed in deionized water containing the metal chelating agent EDTA-2Na [2] and the antimicrobial agent isothiazolinone [3] for moisture stabilization, preliminary desalination, and microbial inhibition. At the same time, it is necessary to regularly monitor the water temperature, pH, concentration of main ions, conductivity, and microbial composition of the desalination buffer, and replace the desalination buffer regularly. The above treatment lays a solid foundation for the overall desalination, reinforcement, and protection of the hull in the later stage.

Wooden cultural relics are a carrier of ancient human civilization and represent a valuable material for the study of ancient history, art, science and technology, economy, and so on. Wooden cultural relics exist in many cultural sites, usually in the form of houses, tombs, hulls, decorations, and so on [4]. Wooden cultural relic can be divided into dry type and waterlogged type. In the original environment, the cellulose in the cell wall of waterlogged wooden cultural relics has been partially or completely degraded by bacteria or fungi. After the loss of degradation products, the structure of the cell wall becomes loose and even produces a large number of holes, and the original place of the cell wall is filled with water. This makes the moisture content of waterlogged archaeological wood (WAW) much higher than ordinary wood [5,6]. Generally, the moisture content of ordinary fresh wood is about 20%, while that of WAW can reach more than 500% [7,8]. The texture of WAW is fragile, so it is difficult to maintain a good condition after excavation. In general, WAW will be preserved in water after being excavated, and the temperature and humidity of the preservation environment will be as stable as possible. One is to maintain the moisture in the wood, so as to maintain the waterlogged state of the wooden cultural relic; the other is to cut off the air and avoid the further reproduction of aerobic microorganisms. The maritime wooden cultural relic is a kind of classic WAW. Due to long-term immersion in seawater, the intracellular electrolytes of maritime WAW have reached a full balance with seawater, and the wood contains a lot of salt. After the maritime WAW is excavated, many physical and chemical reactions will occur due to the change of environmental temperature and humidity, leading to the corrosion of wood. A high concentration of salts can degrade the fibers in WAW. When the environmental temperature and humidity change, some salts may crystallize and dissolve repeatedly, which will lead to fiber degradation and fracture. In addition, there are insoluble salts such as sulfur iron compounds in maritime WAW that easily oxidize into sulfuric acid during long-term exposure, leading to wood acidification and corrosion [2,9]. For example, in the protection of the Swedish warship Vasa, the problem of the acidity of wood and iron compounds in wood was highly apparent [10–12]. Research on the impact of biological pathways of iron and sulfur oxidization on the protection of the Mary Rose was also being studied [13]. At the same time, due to the large volume of maritime WAW, the desalination process is very slow. Therefore, it is necessary to carry out a long-term desalination treatment for maritime WAW to reduce its salt content.

During the preservation of maritime WAW, a variety of disease problems may occur. Therefore, it is necessary to regularly monitor the properties of wood, the nature of the desalination buffer, and microbial biodegradation, and replace the desalination buffer regularly. Once maritime WAW is excavated, it will come into contact with the surrounding environment, which may cause biodegradation problems. Biodegradation is mainly caused by microbial activities, mainly including bacteria and fungi, whose rapid growth and

secretion of secondary metabolites may lead to damage to cultural relics [14]. Wood is an excellent organic substrate for the growth of microorganisms [15]. There are many bacteria and fungi that can produce cellulolytic enzymes and ligninolytic enzymes [16–18]. Some researchers analyzed the bacterial community in 108 samples of WAW from different ages and identified a variety of bacteria [19]. The main disease microorganisms of the "Xiaobaijiao No. 1" shipwreck were erosion bacteria (EB) and tunneling bacteria (TB) [15]. EB and soft rot fungi were found to be active in WAW [20]. Therefore, it is necessary to comprehensively consider wood properties and microbial problems, in order to better solve various diseases of WAW.

## 2. Materials and Methods

### 2.1. Sample Collection

In this study, we mainly monitored monitoring tank 8 (NH.W2) in the archaeological excavation site of the Nanhai No. 1 shipwreck. The average annual temperature of the site is 25.6 °C and the average annual humidity is 84.1% [21]. The NH.W2 had a capacity of one cubic meter, and contained about 20 pieces of scattered wood of different sizes from the hull. Identified by archaeologists, the hull wood of the Nanhai No. 1 shipwreck was mainly *Pinus latteri* [22]. The scattered wood unearthed during the excavation were washed to remove the surface sea mud, and then gradually put into NH.W2 for preliminary desalination. In NH.W2, deionized water with 10mmol/LEDTA-2Na and 0.5% isothiazolinone K100 was used as the desalination buffer. The wood properties of the WAW in NH.W2 were detected in April and July 2019. The iron content of the desalination buffer in NH.W2 was detected in August and September 2019. The anion and cation content of the desalination buffer in NH.W2 was detected in May, June and October 2019. The microbial composition of the desalination buffer in NH.W2 was detected in November 2019 (Figure 1). In November 2019, the pH of the desalination buffer in NH.W2 was 4.83, and the water temperature was 20.8 °C. The desalination buffer was collected by 50 mL aseptic centrifuge tubes, then cryopreserved and transported to the laboratory.

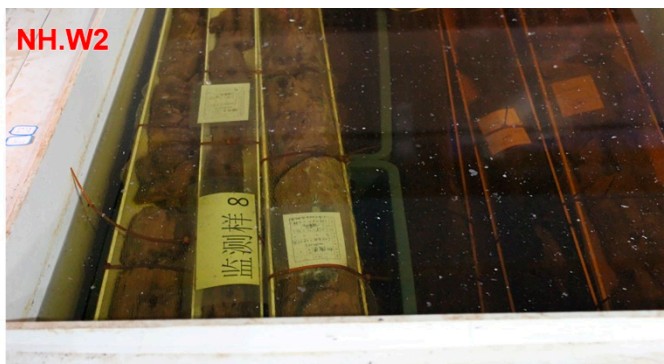

**Figure 1.** Sampling picture of monitoring tank 8 (NH.W2) in November 2019.

### 2.2. Detection of Chemical Components in the Wood

The same piece of wood in NH.W2 was sampled in April and July 2019 to detect chemical components (including lignin, holocellulose, and ash content) in the wood. Then, the degradation degree and ash content of the WAW were evaluated by comparing with the content of lignin, holocellulose and ash in fresh *Pinus* wood. The content of lignin in wood was determined according to the method specified in the national standard GB/T 2677.8-94 (fibrous raw material—determination of acid-insoluble lignin) [23]. The content of holocellulose in wood was determined according to the method specified in national standard GB/T 2677.10-1995 (fibrous raw material—determination of holocellulose) [24]. The ash content of wood was determined according to the method specified in the national standard GB/T 2677.3-1993 (fibrous raw material—determination of ash) [25]. GB/T refers to the national standard of the people's Republic of China.

### 2.3. Detection of Iron Element Content in the Buffer

The iron content of the desalination buffer in NH.W2 was detected in August and September 2019 to evaluate the removal effect of insoluble salts mainly composed of sulfur iron compounds. The iron content was detected by inductively coupled plasma mass spectrometer (ICP-MS) (Elan 9000, PerkinElmer, Waltham, MA, USA). Due to the high iron element in the desalination buffer, the sample needed to be pretreated. The water sample was diluted 1000 times with ultrapure water and filtered to prepare 2% $HNO_3$ aqueous solution for testing. The specific parameters were as follows: the scanning was performed 20 times, repeating three times; the sample injection washing time was 45 s with a reading delay time of 15 s, sample residence time of 50 ms in detector, sample injection pump speed of 26 r/min; the sample chamber vacuum was less than $2 \times 10^{-7}$ PA; the protective gas was argon, gas pressure $415 \pm 7$ kPa; the cooling water pressure was 45 to 65 kPa and the temperature was $20 \pm 2$ °C.

### 2.4. Detection of Anion and Cation Content in the Buffer

The anion and cation content of the desalination buffer in NH.W2 were detected in May, June and October 2019 to evaluate the removal effect of soluble salts. The anion and cation ions include $Na^+$, $Cl^-$, and $SO_4^{2-}$. The detection of anion and cation content used ion chromatography (IC) (HIC-10A super IC, Kyoto, Japan). Due to the high anion and cation content in the desalination buffer, the sample needed to be pretreated. The water sample was diluted 100 times with ultrapure water, filtered and injected for determination. The anion test conditions were as follows: eluent: 0.35 mmol/L $Na_2CO_3$ solution, column temperature 45 °C, flow rate 0.8 mL/min, column pressure 11 MPa, injection volume 60 μL. The cation test conditions were as follows: eluent: 0.70 mmol/L $H_2SO_4$ solution, column temperature 40 °C, flow rate 1.0 mL/min, column pressure 3.7 MPa, injection volume 60 μL. The standard samples of $Na^+$, $Cl^-$, and $SO_4^{2-}$ were all 100 mg/L. $Na_2CO_3$ and $H_2SO_4$ were both analytically pure.

### 2.5. Total DNA Extractions and High-Throughput Sequencing

The microbial composition of the desalination buffer in NH.W2 was detected in November 2019 to evaluate the biodegradation problem in the desalination buffer. The water sample total DNA was extracted using DNeasy PowerWater Kit (QIAGEN, Hilden, Germany). The total DNA was sent to Novogene Genome Sequencing Company for high-throughput sequencing. We analyzed the 16SV4 region (bacteria) and ITS1-5F region (fungi) to comprehensively detect the microbial composition in the desalination buffer.

## 3. Results

### 3.1. Analysis of Chemical Components in the Wood

The contents of lignin, holocellulose and ash are shown in Table 1. According to literature reports, in fresh *Pinus* wood, the lignin content is generally about 25%, the holocellulose content is generally about 75%, and the ash content is generally less than 1% [26,27]. Through comparison, it can be found that the holocellulose content was much lower than that of fresh wood, while the ash content was higher than that of fresh wood. Since most of the cellulose had been degraded and lignin is difficult to degrade, the percentage of lignin in the wood has increased to 63%. Therefore, the wood degradation degree of WAW from the Nanhai No. 1 shipwreck was high, and the ash content of the WAW was high. In addition, the changes of lignin and holocellulose contents in April and July 2019 were not obvious, indicating that the degradation of the WAW became slow during the desalination. The ash content in July 2019 was significantly lower than that in April 2019, indicating that some inorganic salt ions were slowly removed after immersion in desalination buffer.

**Table 1.** Lignin, holocellulose, and ash content of WAW in NH.W2 in April and July 2019.

|  | Lignin Content (%) | Holocellulose Content (%) | Ash Content (%) |
|---|---|---|---|
| 20 April 2019 | 61.32 | 4.72 | 13.02 |
| 12 July 2019 | 64.96 | 4.43 | 3.86 |
| Mean Value | 63.14 | 4.575 | 8.44 |

### 3.2. Analysis of Iron Element Content in the Buffer

The iron contents of the desalination buffer are shown in Table 2. We found that the iron content in the desalination buffer increased significantly after one month of the desalination treatment. The results showed that some insoluble salts mainly composed of sulfur iron compounds in WAW could be removed slowly after soaking in desalination treatment.

**Table 2.** Iron element content of the desalination buffer in NH.W2 in August and September 2019.

|  | Iron Element Content (g/L) |
|---|---|
| 1 August 2019 | 127.23 |
| 10 September 2019 | 611.47 |

### 3.3. Analysis of Anion and Cation Content in the Buffer

The anion and cation content of the desalination buffer are shown in Table 3. In the original desalination buffer, $Cl^-$ and $SO_4^{2-}$ did not exist. The $Na^+$ in the original desalination buffer mainly existed in EDTA-2Na, and the content of original $Na^+$ was 460 mg/L. We found that after immersion in the desalination buffer, the contents of $Na^+$, $Cl^-$ and $SO_4^{2-}$ in the desalination buffer were significantly higher than those in the original desalination buffer. Moreover, the contents of $Na^+$, $Cl^-$ and $SO_4^{2-}$ tended to be stable after a period of desalination treatment. The results showed that the soluble salt in WAW was gradually removed after immersion in the desalination buffer.

**Table 3.** Anion and cation content of the desalination buffer in NH.W2 in May, June and October 2019.

|  | $Na^+$ Content (mg/L) | $Cl^-$ Content (mg/L) | $SO_4^{2-}$ Content (mg/L) |
|---|---|---|---|
| 29 May 2019 | 786.30 | 386.12 | 162.23 |
| 28 June 2019 | 745.33 | 512.67 | 219.22 |
| 22 October 2019 | 786.30 | 396.81 | 170.95 |

### 3.4. Microbial Diversity Analysis by High-Throughput Sequencing

We detected the microbial composition of the desalination buffer in NH.W2 in November 2019. Figure 2a,b show the composition and proportion of bacteria in NH.W2 desalination buffer. Figure 2a represents the distribution of the bacteria at the phylum level. The results show that at the phylum level, Proteobacteria accounts for the largest proportion, accounting for 71.95%. This is followed by Firmicutes and Actinobacteria, accounting for 13.90% and 10.04%, respectively. Figure 2b and Table 4 represent the distribution of the bacteria at the genus level. We found that at the genus level, the most abundant bacteria is *Pseudomonas*, accounting for 34.06%. Followed by *Methylobacillus* and *Pandoraea*, accounting for 11.33% and 7.25%, respectively. In addition, there are unidentified *Mitochondria*, *Microbacterium*, *Agromyces*, *Leifsonia*, and *Escherichia-Shigella*. Figure 2c,d show the composition and proportion of fungi in NH.W2 desalination buffer. Figure 2c represents the distribution of the fungi at the phylum level. The results show that at the phylum level, Basidiomycota accounts for the largest proportion, accounting for 99.63%, followed by Ascomycota, accounting for 0.25%. Figure 2d and Table 4 represent the distribution of the fungi at the genus level. We found that at the genus level, the most abundant fungi is *Cutaneotrichosporon*, accounting for 99.59%, followed by *Fusarium* and *Cryptococcus*, accounting for 0.18% and 0.03%, respectively.

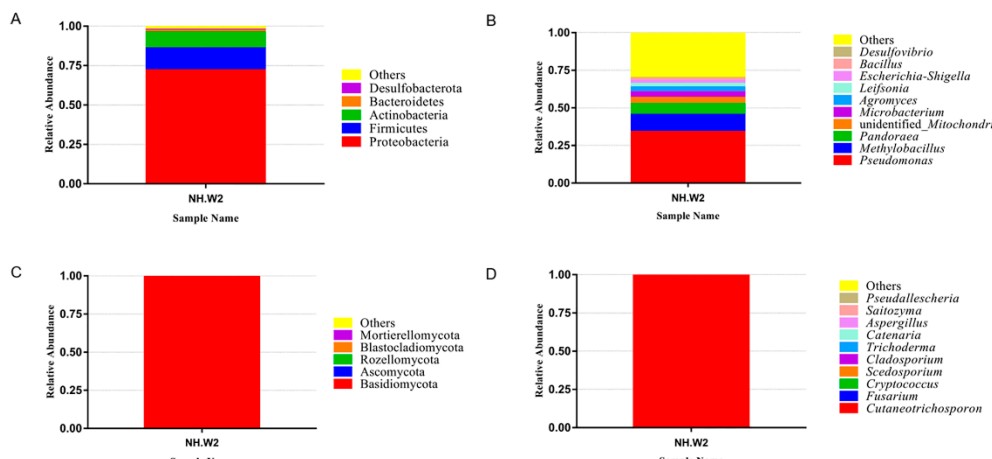

**Figure 2.** The relative abundance of microbial communities in monitoring tank 8 (NH. W2). The relative abundance is shown as a percentage. Phylum and genera are colored according to the legend on the right. (**A**) Relative abundance of bacteria at phylum level. (**B**) Relative abundance of bacteria at genera level. (**C**) Relative abundance of fungi at phylum level. (**D**) Relative abundance of fungi at genera level.

**Table 4.** Relative abundance of dominant bacteria and dominant fungi of the desalination buffer in the NH.W2 at the genus level.

| Dominant Bacteria Genus (%) | | Dominant Fungi Genus (%) | |
| --- | --- | --- | --- |
| *Pseudomonas* | 34.06 | *Cutaneotrichosporon* | 99.59 |
| *Methylobacillus* | 11.33 | *Fusarium* | 0.18 |
| *Pandoraea* | 7.25 | *Cryptococcus* | 0.03 |
| unidentified_*Mitochondria* | 3.99 | *Scedosporium* | 0.01 |
| *Microbacterium* | 3.75 | *Cladosporium* | 0.01 |
| *Agromyces* | 3.12 | *Trichoderma* | 0.01 |
| *Leifsonia* | 2.53 | *Catenaria* | 0.01 |
| *Escherichia-Shigella* | 2.09 | *Aspergillus* | 0.01 |
| *Bacillus* | 1.18 | *Saitozyma* | 0.01 |
| *Desulfovibrio* | 0.42 | *Pseudallescheria* | 0.00 |
| Others | 30.28 | Others | 0.16 |

## 4. Discussion

The Nanhai No. 1 shipwreck is a complex organism, and its excavation and protection represent a great challenge. In this study, we monitored monitoring tank 8 (NH.W2), where the scattered wood was stored in the archaeological excavation site of the Nanhai No. 1 shipwreck. It mainly included the chemical components (including lignin, holocellulose, and ash content) of the wood, the iron element content, anion and cation content in the desalination buffer, and the microbial composition in the desalination buffer. Through the analysis of the above data, a preliminary evaluation is made on the preservation status of WAW from the Nanhai No. 1 shipwreck, which provides data support for the protection of the No. 1 shipwreck in the later stage.

Through the detection, we have drawn the following conclusions: (1) compared with fresh wood, the degradation degree of cellulose in WAW was higher, and the ash content in WAW was higher. The degradation of WAW became slow during the desalination treatment. After immersion in desalination buffer, some inorganic salt ions in WAW were removed slowly. (2) After immersion in the desalination buffer, the iron content in the desalination buffer increased significantly. It showed that some insoluble salts mainly composed of iron compounds in WAW could be removed slowly after soaking in desalination treatment. (3) After immersion in the desalination buffer, the contents of $Na^+$, $Cl^-$ and $SO_4^{2-}$ in the desalination buffer were significantly higher than those in the original desalination buffer.

Moreover, the contents of Na$^+$, Cl$^-$ and SO$_4^{2-}$ tended to be stable after a period of the desalination treatment. The results showed that the soluble salt in WAW was gradually removed after immersion in desalination buffer. (4) The high-throughput sequencing results of the desalination buffer showed that at the bacteria phylum level, Proteobacteria accounts for the largest proportion, accounting for 71.95%. At the bacteria genus level, the most abundant bacteria is *Pseudomonas*, accounting for 34.06%. At the fungi phylum level, Basidiomycota accounts for the largest proportion, accounting for 99.63%. At the fungi genus level, the most abundant fungi is *Cutaneotrichosporon*, accounting for 99.59%.

At the initial stage of the excavation of the Nanhai No. 1 shipwreck, archaeologists carried out a preliminary test on the wood properties of the Nanhai No. 1 shipwreck. The Nanhai No. 1 shipwreck used different species of wood in each part of the hull, and it was found that the main wood was *Pinus latteri* [22]. The average moisture content of wood was 300 to 700%, which indicated moderate and severe corrosion. After analyzing the chemical composition of the wood, it was found that the content of cellulose and hemicellulose in the wood was much lower than that in fresh wood, which indicated that the degree of decay of the wood was high. The ash content in the wood was much higher than that in normal wood, indicating that the salt content in the wood was higher. The salt composition in the hull wood was analyzed, and it was found that there were soluble salts mainly composed of NaCl and insoluble salts mainly composed of sulfur iron compounds, and the content of Na$^+$, Cl$^-$ and SO$_4^{2-}$ were high. The pH of hull wood was between 6 and 7, which belonged to neutral and weak acidity. Combined with the results of this research, we can preliminarily evaluate the preservation status of WAW from the Nanhai No. 1 shipwreck. By storing the WAW in the desalination buffer, the progress of wood degradation may be slowed down to a certain extent, and most of the soluble salt and a small part of the slightly soluble salt can be removed slowly.

In this study, we detected the microbial composition in the desalination buffer. The pH of the desalination buffer was 4.83 and the water temperature was 20.8 °C. *Pseudomonas* and *Cutaneotrichosporon* were the most abundant bacteria and fungi in NH.W2 desalination buffer, respectively. *Pseudomonas* is a common bacterium; it likes to live in humid environments, it grows well in acidic environments [28], and has a high salt tolerance [29]. *Pseudomonas* may participate in the metabolism of sulfur and iron [30,31] and has the ability to degrade cellulose and lignin [32]. *Cutaneotrichosporon* is a kind of yeast with a wide living environment [33] and is often isolated from clinical samples [34]. *Cutaneotrichosporon* has a certain resistance to antifungal drugs, and has a certain acid resistance, high salt tolerance and metal tolerance [35–37]. The ability of microorganisms to destroy wood is determined by a variety of parameters, including the interaction between various organisms, as well as environmental temperature and humidity and other factors [38–40]. Therefore, the biodegradation problem of wooden cultural relics needs to be paid great attention.

## 5. Conclusions

To sum up, the preservation status of WAW from the Nanhai No. 1 shipwreck was comprehensively analyzed and evaluated through the detection of the properties of WAW and the detection of the desalination buffer. This study provides data support for the overall protection of the Nanhai No. 1 shipwreck in the later stage and provides a reference for the protection of other WAW.

**Author Contributions:** Conceptualization, J.P.; Data curation, J.D., X.H., K.M., Y.W. and P.G.; Resources, N.L. and Z.Z.; Writing—original draft, Y.H.; Writing—review and editing, J.P. All authors have read and agreed to the published version of the manuscript.

**Funding:** This research was funded by Natural Science Foundation of Tianjin (19JCZDJC33700) and National Key R&D Program of China [2020YFC1521800].

**Institutional Review Board Statement:** Not applicable.

**Informed Consent Statement:** Not applicable.

**Data Availability Statement:** The raw high-throughput sequencing data can be downloaded at the NCBI Sequence Read Archive (SRA) with the study accession number PRJNA721031.

**Acknowledgments:** We gratefully acknowledge the assistance of Dawa Shen from Chinese Academy of Cultural Heritage.

**Conflicts of Interest:** The authors declare no conflict of interest. The funders had no role in the design of the study; in the collection, analyses, or interpretation of data; in the writing of the manuscript, or in the decision to publish the results.

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
