# Peer review of "Chemical Properties and Microbial Analysis of Waterlogged Archaeological Wood from the Nanhai No. 1 Shipwreck"

_forests, doi:10.3390/f12050587_

Round 1

Reviewer 1 Report

Title:

Properties and Microbial Analysis of Waterlogged Archaeological Wood from the Nanhai No. 1 Shipwreck

Author: Yeqing Han  et al.

General

The article describes part of a study in which the authors analyzed the wood of a sunken ship that had been in salt water for a very long time. In the article, the authors analyze the composition of the solution in which the found wood was stored when the wood was excavated. The amount of different anions and cations in the solution was determined. The authors also analyzed the amount of cellulose, lignin and ash in the submerged wood and also identified the fungi present in the submerged wood.

The article is interesting because it provides insight into the composition, or change in composition, of wood that has been exposed to salt water for centuries, as well as various components that have accumulated in the wood over time. Nevertheless, the article has some minor flaws.

General comments

In most parts of the article, the authors use the active way of writing, ie the first person plural, although scientific articles are usually written in the passive form.

Specific comments

 L2 It is not clear from the title that the research analyzes the chemical components found in the wood and the composition of the wood. The present title is expected to focus more on the properties of wood. Therefore, I suggest changing the address, like: »Chemical Properties and Microbial Analysis of Waterlogged Archaeological Wood from the Nanhai No. 1 Shipwreck« or similar…

L34 The Nanhai No. 1 shipwreck was a wooden merchant ship of the Southern Song Dynasty, which sank in the South China Sea in Guangdong Province, China. – In which year (century) did the ship sink?

L71-72 Generally, the moisture content of ordinary wood is about 20%,… -  Which ordinary wood? Fresh, air dried?

L107 -108 In this study, we mainly monitored the monitoring tank 8 (NH. W2) where the scattered wood was stored in the archaeological excavation site of the Nanhai No. 1 shipwreck. –

Describe the tank in more detail. How large is it, what quantity of wood was stored in it? How was the wood added? Was the wood added all at once, or gradually? Next, in lines 111 through 115, it is indicated when the properties and composition of the solution were determined. Indicate when the wood was added, or how much time elapsed between the addition of the wood to the tank and the determination of each component in the solution and the composition of the wood. Also indicate what type of wood was analyzed.

L126 Then the degradation degree and ash content of WAW were evaluated by comparing with the content of lignin, holocellulose and ash in fresh wood. – Which fresh wood?

L168-169 The contents of lignin, holocellulose and ash are shown in Table 1. According to literature reports, in fresh pine tree wood… - What is the latin name of the specie?

L176-177 In addition, the changes of lignin and holocellulose contents in April and July 2019 were not obvious… - On what basis can this be claimed? How many samples were analyzed? What is the standard deviation of the value? I suggest doing an Anova test to confirm that there is no significant difference between the values.

L203 Table 3. Why the value of Cl and SO4 first increases and then decreases? Comment on the difference, or why the value does not continue to rise or stabilize at the final value?

L225 Figure 2. Enlarge the labels on the Figure.

L266 The Nanhai No. 1 shipwreck used different species of wood in each part of the hull, and it was found that the main wood was pine tree. – How was the tree species determined and what is the Latin name?

Author Response

Dear Reviewer 1:

Thank you for your comments concerning our manuscript. Those comments are valuable and very helpful for revising and improving our paper, as well as the important guiding significance to our researches. We have studied comments carefully and have made correction which we hope to meet with approval. Below, please find our point-by-point responses to your comments. The revised parts are marked in red in the article.

We look forward to hearing from you regarding our submission. We would be glad to respond to any further questions and comments that you may have.

Sincerely yours,

Yeqing Han

General comments: In most parts of the article, the authors use the active way of writing, ie the first person plural, although scientific articles are usually written in the passive form.

Response: We appreciate your comment. According to your suggestion, we have seriously revised the problem of writing.

Point 1: L2 It is not clear from the title that the research analyzes the chemical components found in the wood and the composition of the wood. The present title is expected to focus more on the properties of wood. Therefore, I suggest changing the address, like: »Chemical Properties and Microbial Analysis of Waterlogged Archaeological Wood from the Nanhai No. 1 Shipwreck« or similar…

Response 1: We appreciate your comment. According to your suggestion, we change the title to “Chemical Properties and Microbial Analysis of Waterlogged Archaeological Wood from the Nanhai No. 1 Shipwreck”. (See page 1, line 2)

Point 2: L34 The Nanhai No. 1 shipwreck was a wooden merchant ship of the Southern Song Dynasty, which sank in the South China Sea in Guangdong Province, China. – In which year (century) did the ship sink?

Response 2: We appreciate your comment. The Nanhai No. 1 shipwreck was sank in the Southern Song Dynasty (1127-1279 AD). We add the “(1127-1279 AD)” in the page 1, line 35.

Point 3: L71-72 Generally, the moisture content of ordinary wood is about 20%,… -  Which ordinary wood? Fresh, air dried?

Response 3: We appreciate your comment. Here we mean ordinary fresh wood, so we change the “ordinary wood” to “ordinary fresh wood”. (See page 2, line 71)

Point 4: L107 -108 In this study, we mainly monitored the monitoring tank 8 (NH. W2) where the scattered wood was stored in the archaeological excavation site of the Nanhai No. 1 shipwreck. –

Describe the tank in more detail. How large is it, what quantity of wood was stored in it? How was the wood added? Was the wood added all at once, or gradually? Next, in lines 111 through 115, it is indicated when the properties and composition of the solution were determined. Indicate when the wood was added, or how much time elapsed between the addition of the wood to the tank and the determination of each component in the solution and the composition of the wood. Also indicate what type of wood was analyzed.

Response 4: We appreciate your comment. According to your suggestion, we add “The NH.W2 had a capacity of one cubic meter, and contained about 20 pieces of scattered wood of different sizes from the hull. Identified by archaeologists, the hull wood of the Nanhai No. 1 shipwreck was mainly Pinus latteri [22]. The scattered wood unearthed during the excavation were washed to remove the surface sea mud, and then gradually put into NH.W2 for preliminary desalination.” to describe the tank in more detail, and added references [22] to demonstrate the main tree species. (See page 3, line 113-118).

  1. Naisheng Li; Yue Chen; Dawa Shen. Study on the protection of the excavation site of the Nanhai No. 1 shipwreck (2014-2016); Science Press: Beijing, China, 2017; pp. 124-133.

In lines 111 through 115, We describe the sampling time of the sample: “The wood properties of the WAW in NH.W2 was detected in April and July 2019. The iron content of desalination buffer in NH.W2 was detected in August and September 2019. The anion and cation content of desalination buffer in NH.W2 was detected in May, June and October 2019. The microbial composition of desalination buffer in NH.W2 was detected in November 2019.”.

Point 5: L126 Then the degradation degree and ash content of WAW were evaluated by comparing with the content of lignin, holocellulose and ash in fresh wood. – Which fresh wood?

Response 5: We appreciate your comment. Identified by archaeologists, the hull wood of the Nanhai No. 1 shipwreck was mainly Pinus latteri. We change the “in fresh wood” to “in fresh Pinus wood”. (See page 3, line 134)

Point 6: L168-169 The contents of lignin, holocellulose and ash are shown in Table 1. According to literature reports, in fresh pine tree wood… - What is the latin name of the specie?

Response 6: We appreciate your comment. We change the “in fresh wood” to “in fresh Pinus wood”. (See page 4, line 177)

Point 7: L176-177 In addition, the changes of lignin and holocellulose contents in April and July 2019 were not obvious… - On what basis can this be claimed? How many samples were analyzed? What is the standard deviation of the value? I suggest doing an Anova test to confirm that there is no significant difference between the values.

Response 7: We appreciate your comment. Through the detection of the same piece of wood, we found that the content of lignin and holocellulose did not change significantly in April 2019 and July 2019. Lignin was maintained at 60% and holocellulose at 4%. So we have come to the above conclusion. Unfortunately, at that time, because of the importance of cultural relics, we only selected one piece of wood for testing, and did not collect more samples. So the amount of sample is not enough for standard deviation and ANOVA test.

Point 8: L203 Table 3. Why the value of Cl and SO4 first increases and then decreases? Comment on the difference, or why the value does not continue to rise or stabilize at the final value?

Response 8: We appreciate your comment. The monitoring tank 8 (NH.W2) has a large capacity, the distribution of ion content may be uneven, and the sampling is random. This may lead to the fluctuation of ion content. So, on the whole, Na+, Cl- and SO42- tended to be stable after a period of desalination treatment. Although the detection of ion concentration is random, the main purpose of ion concentration monitoring in the archaeological site is to get a general understanding of desalination situation. When the monitoring personnel find that the ion concentration has basically reached a stable state, the desalination solution will be replaced in time.

Point 9: L225 Figure 2. Enlarge the labels on the Figure.

Response 9: We appreciate your comment. We have enlarged the labels on the Figure 2. (See page 6, line 232)

Point 10: L266 The Nanhai No. 1 shipwreck used different species of wood in each part of the hull, and it was found that the main wood was pine tree. – How was the tree species determined and what is the Latin name?

Response 10: We appreciate your comment. Identified by archaeologists, the hull wood of the Nanhai No. 1 shipwreck was mainly Pinus latteri. We change the “pine tree” to “Pinus latteri”. (See page 7, line 274)

Thank you!

Reviewer 2 Report

It is always nice to have a contribution on historical shipwrecks reported, and this paper provides relevant data on the microbial degradation of a Chinese historical piece.

I have the following recommendations before the article is accepted:

  • Please try and structure it better by clearly individuating your starting hypothesis, methodological approach and conclusions.
  • Since this is a nice conservation case, it would be important to relate it, somewhere between the introduction and the discussion, to other shipwreck conservation cases, where either iron contamination or acidic species brought on degradation after microbial attack, namely the Vasa and the Mary Rose. Here are some possible references that should help the author put their work in context:

“The Effects of Mary Rose Conservation Treatment on Iron Oxidation Processes and Microbial Communities Contributing to Acid Production in Marine Archaeological Timbers”, Plos One 2014, 9.

“Nanotechnology for Vasa wood de-acidification”, Macromolecular Symposia 238, 2006, Pages 30-36

“Conservation of acid waterlogged shipwrecks: Nanotechnologies for de-acidification” Applied Physics A: Materials Science and Processing, 83, 2006, Pages 567-571

“Extraction of iron compounds from wood from the Vasa”, 2006 Holzforschung 60(6): 678-684

  • Please improve readability of figure 2, and report errors in Table 4.

Author Response

Dear Reviewer 2:

Thank you for your comments concerning our manuscript. Those comments are valuable and very helpful for revising and improving our paper, as well as the important guiding significance to our researches. We have studied comments carefully and have made correction which we hope to meet with approval. Below, please find our point-by-point responses to your comments. The revised parts are marked in red in the article.

We look forward to hearing from you regarding our submission. We would be glad to respond to any further questions and comments that you may have.

Sincerely yours,

Yeqing Han

Point 1: Since this is a nice conservation case, it would be important to relate it, somewhere between the introduction and the discussion, to other shipwreck conservation cases, where either iron contamination or acidic species brought on degradation after microbial attack, namely the Vasa and the Mary Rose. Here are some possible references that should help the author put their work in context:

“The Effects of Mary Rose Conservation Treatment on Iron Oxidation Processes and Microbial Communities Contributing to Acid Production in Marine Archaeological Timbers”, Plos One 2014, 9.

“Nanotechnology for Vasa wood de-acidification”, Macromolecular Symposia 238, 2006, Pages 30-36

“Conservation of acid waterlogged shipwrecks: Nanotechnologies for de-acidification” Applied Physics A: Materials Science and Processing, 83, 2006, Pages 567-571

“Extraction of iron compounds from wood from the Vasa”, 2006 Holzforschung 60(6): 678-684

Response 1: We appreciate your comment. The Vasa and the Mary Rose are good cases to describe iron contamination and acidic degradation of WAW. According to your suggestion, we added “For example, in the protection of the Swedish warship Vasa, the problem of the acidity of wood and iron compounds in wood had been highly valued [10-12]. Research on the impact of biological pathways of iron and sulfur oxidization on the protection of the Mary Rose was also being studied [13].”, and added references [10-13]. (See page 2, line 87-90)

  1. Chelazzi D; Giorgi R; Baglioni P. Nanotechnology for Vasa Wood De-Acidification. Macromolecular Symposia 2010, 238(1), 30-36. DOI: 10.1002/masy.200650605.
  2. Giorgi R; Chelazzi D; Baglioni P. Conservation of acid waterlogged shipwrecks: nanotechnologies for de-acidification. Applied Physics A 2006, 83(4), 567-571. DOI: 10.1007/s00339-006-3542-z.
  3. Almkvist G; Persson I. Extraction of iron compounds from wood from the Vasa. Holzforschung 2006, 66(6), 1125-684. DOI: 10.1515/HF.2006.114.
  4. Joanne P; Smith A D; Schofield E J; Chadwick A V; Jones M A; Watts J E M. The Effects of Mary Rose Conservation Treatment on Iron Oxidation Processes and Microbial Communities Contributing to Acid Production in Marine Archaeological Timbers. PLoS ONE 2014, 9(2), 1-8. DOI: 10.1371/journal.pone.0084169.

Point 2: Please improve readability of figure 2, and report errors in Table 4.

Response 2: We appreciate your comment. We have enlarged the labels on the Figure 2, and fixed the errors in the Table 4 (in red). (See page 6, line 232 and page 6, line 241)